# Enrichment of Brain n-3 Docosapentaenoic Acid (DPA) and Retinal n-3 Eicosapentaenoic Acid (EPA) in Lambs Fed *Nannochloropsis oceanica* Microalga

**DOI:** 10.3390/ani13050828

**Published:** 2023-02-24

**Authors:** Ana C. M. Vítor, Jorge J. Correia, Susana P. Alves, Rui J. B. Bessa

**Affiliations:** 1Faculdade de Medicina Veterinária, Universidade de Lisboa, Avenida da Universidade Técnica, 1300-477 Lisboa, Portugal; 2CIISA—Centro de Investigação Interdisciplinar em Sanidade Animal, Avenida da Universidade Técnica, 1300-477 Lisboa, Portugal; 3Associate Laboratory for Animal and Veterinary Sciences (AL4AnimalS), Avenida da Universidade Técnica, 1300-477 Lisboa, Portugal

**Keywords:** fatty acid, hippocampus, prefrontal cortex, dimethyl acetal

## Abstract

**Simple Summary:**

Omega-3 polyunsaturated fatty acids (n-3 PUFAs), mainly eicosapentaenoic acid (EPA) and docosahexaenoic acid (DHA), have been increasingly studied due to their beneficial health effects. N-3 PUFAs are particularly abundant in the brain and retina, where they play various roles that are important to the maintenance of normal function in those organs. The present study aimed to evaluate the FA profile of lamb brain and retinal tissues after they were fed three experimental diets supplemented with an EPA-rich microalga for 21 days. The microalga was delivered in a different format in each one of the diets (oil, spray-dried and freeze-dried biomass); therefore, its efficiency in altering the FA profile of brain and retina was evaluated for each diet. Overall, our results demonstrated that the brain EPA content remained unchanged after EPA supplementation, in contrast with the retinal EPA, which was very responsive to microalga supplementation.

**Abstract:**

Omega-3 polyunsaturated fatty acids (n-3 PUFAs) have special physiological functions in both brain and retinal tissues that are related to the modulation of inflammatory processes and direct effects on neuronal membrane fluidity, impacting mental and visual health. Among them, the long-chain (LC) n-3 PUFAs, as eicosapentaenoic acid (EPA) and docosahexaenoic acid (DHA), are of special importance. Scarce data are available about the fatty acid (FA) composition of the ruminant brain in response to dietary intervention. However, we decided to examine the brain and retina FA composition of lambs supplemented with an EPA-rich microalga feed for 21 days, as it is known that despite the extensive biohydrogenation of dietary PUFAs in the rumen, ruminants can selectively accumulate some n-3 LC-PUFAs in their brain and retinal tissues. Twenty-eight male lambs were fed a control diet, or the same diet further supplemented with *Nannochloropsis* sp. microalga. Their brains and retina were collected for FA characterization. Overall, the brain FA profile remained unchanged, with little alteration in omega-3 docosapentaenoic acid (DPA) enhancement in both the hippocampus and prefrontal cortex. Retinal tissues were particularly responsive to the dietary intervention, with a 4.5-fold enhancement of EPA in the freeze-dried-fed lambs compared with the control lambs. We conclude that retinal tissues are sensitive to short-term n-3 PUFA supplementation in lambs.

## 1. Introduction

Fatty acids (FAs) are the most abundant organic compounds in the brain. More than 90% of polyunsaturated FAs (PUFAs) in the mammalian brain are composed of long-chain PUFAs (i.e., ≥20 C chain, LC-PUFA) such as arachidonic acid (AA, 20:4n-6) and docosahexaenoic acid (DHA, 22:6n-3) [1]. Most of these PUFAs are esterified in brain membrane phospholipids (PLs), and they influence brain functions by altering the biophysical properties of cell membranes [2]. Omega-3 PUFAs (n-3 PUFAs) have a special importance since they play a role in a variety of physiological functions related to neurogenesis, neurotransmission, and neuroinflammation, contributing to the development, functioning, and ageing of the brain. Furthermore, in humans, n-3 LC-PUFA dietary deficiencies are associated with an increased risk of developing various psychiatric disorders [3] and neurodegenerative diseases [4]. Among these FAs, eicosapentaenoic acid (EPA, 20:5n-3) and DHA have been linked to the maintenance of mental health, mediated by the modulation of inflammatory processes and direct effects on neuronal membrane fluidity and receptor function [3]. N-3 docosapentaenoic acid (DPA, 22:5n-3) also plays an important role, as it is the second most abundant n-3 LC-PUFA in the brain after DHA. It is suggested to be specifically beneficial for elderly neuroprotection and early-life brain development [5].

In addition to the brain, retinal tissue also is rich in lipids, which comprise approximately 20% of the retina’s dry weight. Retinal membrane PLs have the highest level of LC-PUFAs of any tissue in humans (approximately 33%) [6,7,8]. The retina is a tissue with a naturally high content of n-3, particularly DHA, which plays an essential role in optimizing the fluidity of photoreceptor membranes, retinal integrity and visual function [9]. DHA also has a protective role in the retina, participating in the anti-inflammatory activity, anti-angiogenesis, anti-apoptosis and providing protection from neurotoxicity [9].

The brain lipids of ruminant species, especially cattle, have been well characterized and appear to be very similar to those of the human brain in terms of both content and composition [10]. Although they are less well-characterized, the lipids in the brains of sheep also do not seem to differ markedly from the ones found in cattle [10]. In ruminants, brain characterization has been traditionally focused on the types of lipids and less so on a deeper characterization of FAs. Moreover, lipid analyses have been performed primarily in brain homogenates or gross anatomical structures within the brain [10], not in specific functional regions or tracts of the ruminant brain. Bovine and ovine retinal fatty acid compositions are similar: they are composed of appreciable amounts of palmitic acid (16:0), stearic acid (18:0), DHA and AA [10,11]. Thus, despite the extensive biohydrogenation of dietary PUFAs in the rumen, ruminants can selectively accumulate n-3 LC-PUFA in their brain and retinal tissues, contrasting with the low deposition of these FAs in adipose tissue and muscle [12]. Liver stores of n-3 LC-FA were reported to be the primary source of these FAs for the brain tissues of rats [13], even during periods of low dietary intake of these FAs [13,14]. This makes it possible to maintain adequate levels of DHA and AA in the brains of DHA- and AA-deprived animals [13,14]. In a very recent publication, cattle supplemented with calcium salts of fish oil with 11% EPA and 8% DHA had, across most brain regions, greater EPA concentrations when compared to palm-oil-supplemented animals [15].

The dietary supplementation of microalgae has been shown to enhance n-3 LC-PUFAs along lambs’ gastrointestinal tracts [16]. Moreover, Vítor et al. [16] showed that the drying method applied to the microalgae strongly influenced the powder architecture and cell wall integrity, consequently affecting the degree of EPA protection against rumen microbes. Therefore, we hypothesise that lambs’ brains and/or retinal tissues would be sensitive to the differences in n-3 LC-PUFA absorption due to the changes in rumen biohydrogenation associated with the processing of microalgae biomass. Thus, we collected brain and retina samples from six-month-old lambs used in a previous experiment and conducted a detailed FA composition of those tissues.

## 2. Materials and Methods

### 2.1. Animal Handling and Diets

The current lamb trial was conducted in compliance with the ARRIVE and international guidelines. The trial was conducted in certified facilities and was approved by an ethical and animal well-being commission, as fully detailed in Vítor et al. [16]. Twenty-eight sixty-day-old Merino Branco ram lambs with an average body weight of 21.8 ± 4.4 kg were housed in INIAV facilities in Santarém, Portugal. The animals were randomly allocated to individual pens (1.52 m^2^) with ad libitum access to clean water. The lambs were sorted into four experimental groups with seven replicates per group. The experimental diets included a control diet (C diet), consisting of pellets containing dehydrated lucerne, barley and soybean meal and no added sources of EPA, and three diets supplemented with the microalga *Nannochloropsis* sp., designed to provide approximately 3 g of EPA per kg of diet dry matter (DM). The average content of EPA (mg EPA/g product) in each microalgal format was 235 in the *Nannochloropsis* oil, 22.7 in the spray-dried *Nannochloropsis oceanica* and 30.8 in the lyophilized *Nannochloropsis oceanica*. *Nannochloropsis* sp.-containing diets were composed of the C diet plus 123 g/kg of spray-dried *Nannochloropsis oceanica* biomass (SD diet); 92 g/kg freeze-dried *Nannochloropsis oceanica* biomass (FD diet); and 12 g/kg of *Nannochloropsis* sp. free-oil (O diet) (Table 1). The trial had a 3 week duration limitation due to the high cost of the spray-dried *Nannochloropsis oceanica* biomass and the difficulty of obtaining enough freeze-dried biomass with lab-scale equipment.

### 2.2. Slaughter and Sample Collection

After the end of the third week, the animals were slaughtered using a captive bolt. This was followed by exsanguination. The brain tissue was removed whole. It retained its shape and landmarks in spite of the captive bolt damage. The brain was cut on a sagittal plane and divided into two hemispheres, which were then frozen at −80 °C. After thawing, the different parts were individualized from the right hemisphere (Figure 1), stored in individual bags, frozen at −80 °C and lyophilized. Grey and white matter were collected from two different points in the brain and were considered samples from non-function-specific brain parts, representing only samples from the two histologic and physiological brain areas. The prefrontal cortex (cerebral cortex covering the front part of the frontal lobe), and hippocampus (located in the medial part of the temporal lobe and, on a mid-sagittal section of the brain, posterior to the amygdala extending posteriorly to the splenium of the corpus callosum) were selected as function-specific brain parts.

Immediately after slaughter, the right eyeball of each lamb was removed with a spatula, stored in a bag and frozen at −80 °C. The eyeballs were thawed, and the retina and *tapetum lucidum* (RTL) were individualised (Figure 2). The liver was removed from the carcass, and a portion of the left lobe was stored in a bag and frozen at −80 °C. All brain parts, RTL and a portion of the liver were lyophilised prior to fatty acid extraction. 

### 2.3. Fatty Acid Methyl Esters (FAMEs) and Dimethyl Acetals (DMAs) Analysis

Fatty acid methyl esters (FAMEs) and dimethyl acetals (DMAs) of the brain, RTL tissues and liver samples were prepared by acid-catalysed transesterification in methanol [17]. In plasmalogens, the sn-1 position of the glycerol contains a vinyl-ether that releases a DMA in the presence of acid methanol solution. Briefly, approximately 100 mg of lyophilised and ground sample was weighted to reaction tubes. Toluene and 1 mL of internal standard (methyl nonadecanoate −1 mg/mL) were then added, and the samples were placed in an ultrasound bath for ten minutes. A solution of 1.25 M HCl in methanol (3 mL) was added and left to react overnight at 50 °C in a water bath. 

Fatty acid methyl esters and DMAs were analysed by gas chromatography with flame ionization detection (GC-FID), using a Shimadzu GC 2010-Plus (Shimadzu, Kyoto, Japan) equipped with an SP-2560 (100 m × 0.25 mm, 0.20 μm film thickness, Supelco, Bellefonte, PA, USA) capillary column. The injector and detector temperatures were maintained at 220 °C and 250 °C, respectively. The carrier gas was helium at a constant flow of 1 mL/min. The GC oven temperature began at 50 °C for 1 min, then increased to 150 °C at 50 °C/min, held for 20 min, increased to 190 °C at 1 °C/min, and finally increased to 220 °C at 2 °C/min and held for 40 min. The identification of FAMEs was achieved by a comparison of the fatty acid retention times with those of commercial standards (FAME mix, 37 components from Supelco Inc., Bellefont, PA, USA) and with published chromatograms [18,19]. Additional confirmation of FAMEs and DMAs were achieved by electron impact mass spectrometry using a Shimadzu GC–MS QP2010 Plus (Shimadzu, Kyoto, Japan) equipped with a SP-2560 (100 m × 0.25 mm, 0.20 μm film thickness, Supelco, Bellefonte, PA, USA) capillary column and similar GC conditions.

### 2.4. Statistical Analysis

FAME and DMA data were analysed as a completely randomised experimental design using the MIXED procedure of SAS 9.4 (SAS Institute Inc., Cary, NC, USA). Diet was used as a fixed factor, and the animal was used as the experimental unit. Feed intake data were analysed as a completely randomised block design, in which an individual lamb was used as the experimental unit and the model included the treatment and initial live weight block as the fixed factors. The least square means and standard error of the mean (SEM) were reported, and the main effects and their interactions were considered significant at *p*  <  0.05. The TFA + DMA content is presented in mg/g DM, and the FA individual composition is presented in % of TFA + DMA (g FA/100 g TFA + DMA). The sparse partial least squares discriminant analysis (sPLSDA) was performed using MetaboAnalyst 5.0 software, using the centred log ratio transformed FA data as input. 

## 3. Results

### 3.1. Fatty Acid Intake

The feed intake averaged 1.19 ± 0.13 kg (mean ± standard error of the mean) of DM/day during the experiment. It did not differ among treatments (*p* > 0.05) [16]. A brief report of individual and total FA intake (mg/day) is presented in Table 2.

Apart from 16:0, the intake of all the FAs analysed differed between the control and *Nannochloropsis*-diet-supplemented lambs (*p* < 0.05). The FA 14:0, 20:4n-6 and 20:5n-3 were only present in the *Nannochloropsis*-supplemented diets; therefore, their feed intake was zero in the control-fed lambs. The total FA intake differed between control and *Nannochloropsis*-supplemented lambs (*p* < 0.05), being higher in the latter.

### 3.2. Brain Fatty Acid and Dimethyl Acetal Profile

The total FA and DMA (TFA + DMA) content (mg/g DM) and composition (g/100 TFA + DMA) of grey and white matter are presented in Table 3; those of the hippocampus and prefrontal cortex are presented in Table 4. Regarding the grey and white matter, the amount of TFA + DMA did not differ among treatments, averaging 191 and 227 mg/g DM, respectively. Additionally, in both the hippocampus and prefrontal cortex, TFA + DMA content did not differ among treatments, averaging 199 and 208 mg/g DM, respectively.

No major effects of microalgal supplementation (*p* > 0.05) were observed for any of the FAs and DMAs in both the grey and white matter.

Significant differences in the FA composition were observed in the hippocampus and prefrontal cortex (Table 4). In the hippocampus, 20:3n-6 was significantly (*p* < 0.05) lower in the C and SD diets and higher in the O and FD diets. In the prefrontal cortex, this FA was higher in all *Nannochloropsis*-supplemented lambs compared to C-fed lambs. Regarding 22:5n-6: it was significantly higher in the hippocampus in the C-fed lambs when compared to the *Nannochloropsis*-supplemented lambs. In the prefrontal cortex, this FA tended (*p* = 0.056) to follow a similar pattern to what was found in the hippocampus.

DPA had its lowest values (*p* < 0.05) in the hippocampi and prefrontal cortices of C lambs, and its highest values were found in the tissues of lambs fed with the FD supplement. DPA was lower (*p* < 0.05) in these tissues in O-fed lambs than in SD-fed lambs.

Supplement treatments did not affect (*p* > 0.05) the EPA or DHA concentrations in the hippocampus or prefrontal cortex. The sum of EPA + DHA averaged 10% in the hippocampus and 11% in the prefrontal cortex. The total PUFA and n-3 PUFA contents averaged 22% and 12% in both the hippocampus and prefrontal cortex, respectively. 

The content of TFA + DMA was not affected (*p* > 0.05) by dietary treatment. 

### 3.3. Retina and Tapetum Lucidum (RTL)

Similar to what was observed in the brain parts, the TFA + DMA content in the RTL tissues did not differ among treatments, averaging 53 mg/g DM (Table 5). However, more treatment effects occurred for RTL tissues than were observed in brain tissues. 

In RTL tissues, the *c*16-18:1 value was greater (*p* < 0.05) in SD lambs than in all other treatments. Regarding 20:3n-9, a higher content was found in both the C- and SD-fed lambs when compared to the remaining groups. The content of 20:3n-6 was lower in C-fed lambs when compared to the remaining treatments, and 22:4n-6 was higher in both C- and SD-fed lambs when compared to the remaining treatments. Both EPA and DPA were higher in the microalgae-biomass-fed lambs, although there were no differences between the SD- and FD-fed lambs. When compared to the C treatment, biomass-fed lambs had 4.6 times more EPA and twice more DPA in their RTL tissues.

The DHA content in RTL tissues did not differ among treatments (*p* > 0.05).

In the partial sums evaluated, the AA/EPA ratio differed between treatments, being higher in the C-fed lambs when compared to the remaining treatments (*p* < 0.001). The sum of the EPA + DHA averaged 7% TFA. N-3 PUFA averaged 10% ± 1.0, corresponding to approximately 45% of the total PUFAs.

Similar to what was verified in the brain, none of the individual DMAs nor the total DMA content differed between treatments in the RTL tissues. Overall, the total DMA content was much lower than that found in the brain, averaging 4%.

The fold change in the EPA, DPA and DHA content between FD-fed lambs and C-fed lambs was compared between the brain, RTL and liver (Appendix A) and previously analysed subcutaneous adipose tissue (SC AT) and longissimus lumborum muscle samples [20].

The EPA fold change was higher in the SC AT and similar between the liver and RTL. 

### 3.4. sPLSDA Analysis

Figure 3 illustrates a total of five sparse partial least squares-discriminant analysis (sPLS-DA) plots, corresponding to four plots that belong to all the brain parts evaluated (Panels A–D) and one plot belonging to RTL tissues (Panel E). It is possible to observe that there is no clear individualization of lambs belonging to the same diet in accordance with their brain FA and DMA compositions (%TFA) in all brain parts. However, in the RTL tissues (Figure 3E), it is possible to clearly separate C-fed lambs from *Nannochloropsis*-supplemented lambs based on their retinal FA composition.

## 4. Discussion

### 4.1. Brain FA Composition

The classic research on ruminant brain lipids has been based on evaluating the lipid classes and the FAs within the lipid classes of whole brain homogenates [10]. In the present study, we present the FA and DMA profile (expressed as % of TFA + DMA) of the brain and RTL tissues of lambs fed *Nannochloropsis* sp. lipids. 

The most abundant FAs found in the ovine brain were 18:0 (≈18%), 16:0 (≈17%), *c*9-18:1 (≈17%), DHA (≈9%) and AA (≈5% of TFA + DMA). A similar pattern was also observed in the bovine brain regions [15], with the same five FAs also being the most abundant: 16:0 (≈18%), 18:0 (≈20%), *c*9-18:1 (≈24%), AA (≈6%) and DHA (≈9% of total FA). It was shown that linoleic acid (LA), often the major PUFA in muscle, and α-linolenic acid (ALA) were only present at very low levels. In fact, the longissimus lumborum muscles of the same animals presented LA and ALA proportions of circa 9 and 1% [20], contrasting with the proportions of ≈0.5% and ≈0.1% observed in the brain, respectively. Despite being the major dietary PUFA, LA can hardly be considered functional in the brain because of its low concentration (<0.5% of TFA). This low concentration is probably due to LA being extensively converted to AA, which plays a role in neurodevelopment [21]. Moreover, the majority (≈59%) of LA entering the brain is rapidly β-oxidized [22].

We hypothesised that lambs’ brains and/or retinal tissues would be sensitive to the differences in n-3 LC-PUFA absorption due to the changes in rumen biohydrogenation associated with processing the microalgae biomass. Namely, FD *Nannochloropsis oceanica*-supplemented diets appeared to produce a higher n-3 LC-PUFA enhancement in the lambs’ brains when compared to O and SD because freeze-drying better protects the integrity of the microalgal cell wall, reducing the access of ruminal microbiota to the n-3 LC-PUFA inside the cell. In ruminants, almost 90% of dietary lipids reach the duodenum as non-esterified saturated FAs [23], and PUFAs are selectively converted into phospholipid forms in the enterocyte [24]. The transport and uptake of EPA and DHA within brain and retina involves their esterification into a lysophospahtidylcholine and a specific transporter (Mfsd2a) [25,26,27]. Thus we anticipated that the uptake of EPA into the brain and retina would be efficient and responsive to the intestinal absorption of EPA. However, despite the dietary supplementation of EPA, EPA proportions in the brain were low (≈0.6%) and did not differ among treatments in any of the brain parts. This contrasts with the response to EPA deposition observed in the longissimus lumborum muscle of the same animals, in which EPA rose from 0.8% in the C treatment to 1.7% in the *Nannochloropsis*-supplemented treatments [20]. The response in the liver was even more pronounced, with EPA rising from 0.9% up to 4% in the *Nannochloropsis*-supplemented treatments (Appendix A). 

Thus, in general, the ovine brain was not responsive to dietary EPA supplementation. This contrasts with the results reported by Rule et al. [15], in which a similar intake of EPA (2.3 g/day) resulted in an EPA enhancement across various brain parts. However, the supplementation period in our study lasted 1/10 of the one in Rule et al. [15].

As the uptake of EPA and DHA into the brain is similar [28], the lack of EPA enhancement in the lambs’ brains in response to the treatments might be explained by the faster β-oxidation of EPA compared to DHA and/or by the extensive elongation to DPA and subsequent desaturation to DHA. Nevertheless, we also did not observe an increase in DHA. The long half-life of DHA in brain tissues can explain the slow turnover of these fatty acids, therefore explaining the lack of their enhancement in the brain [29]. Moreover, there seems to be evidence that brain DHA and AA levels can be maintained by the liver stores once there is evidence that liver (but not brain) DHA synthesis is upregulated when the dietary content of n-3 PUFA is reduced [13].

Similar responses in brain FAs to microalgae supplementation were observed for the hippocampus and prefrontal cortex, probably reflecting the extensive hippocampal–prefrontal interactions involved in various cognitive and behavioural functions in animals [30,31]. Higher amounts of n-3 DPA and dihomo-γ-linolenic acid (DGLA, 20:3n-6) were found in the hippocampi and prefrontal cortices of microalgae-fed lambs. DGLA is an intermediate of the elongation and desaturation of LA, being converted into AA through the activity of the Δ-5 desaturase enzyme [32]. The increase in DGLA in the brains of lambs supplemented with microalgae was not obvious, as *Nannochloropsis* does not contain relevant amounts of LA and DGLA. DPA, a product of the elongation of EPA, was increased despite the lack of response in EPA and DHA. As it has been proposed that DPA constitutes a storage depot for EPA and DHA [33], its enhancement seems desirable. Although DPA was approximately 10 times lower than DHA in the brain, it was more responsive to the dietary supply of n-3 PUFA. The same pattern was also observed in the brains of lambs suckling from ewes fed with linseed [34] and in the hippocampus of bovines fed fish oil [15].

Most of the beneficial effects of marine oils (mainly fish oils) have been attributed to DHA and EPA [5,35]. However, DPA, which is the intermediate between EPA and DHA in the n-3 LC-PUFA biosynthetic pathway, also presents beneficial biological effects. It reduces platelet aggregation, improves the lipid plasmatic profile, neural health and endothelial cell migration, and assists in the resolution of chronic inflammation [5]. 

Plasmalogens are a subclass of glycerophospholipids that comprise part of biological membranes, including the plasma membrane and the membranes of intracellular organelles, affecting their biophysical properties. They are quantitively important in membranes of neuronal tissues, including the brain and the retina, and are associated with neurological and psychiatric disorders or are involved in the regulation of retinal vascular development, respectively [36,37]. In the DMA, the backbone at the sn-2 position is mainly bonded to PUFAs such as DHA and AA, suggesting its protective role against lipoxidation [38,39]. The DMA content of ruminant brains is not often reported [40]. In our study, the high abundance of plasmalogens in the brain can be perceived through the high DMA content (≈10% of TFA + DMA). The average content of DMA was higher in the brain when compared to the retina (≈3.5% of TFA + DMA). 

### 4.2. RTL Tissue FA Composition

The retina is a thin, highly organised neural tissue lining the posterior aspect of the eye. It is responsible for initiating vision by transducing light into neural signals [41]. The visual streak area of the retina is a narrow horizontal band. It runs parallel to the ventral edge of the *tapetum* [42]. Therefore, due to anatomical proximity and for practical reasons, we collected both tissues simultaneously. The *tapetum lucidum* is a biologic reflector system that is commonly present in the eyes of vertebrates. It enhances visual sensitivity at low light levels by providing light-sensitive retinal cells with a second opportunity for photon–photoreceptor stimulation. Ovine *tapetum lucidum* belongs to the choroidal fibrous type, and the reflective material is made of collagen [43] that constitutes 65% of the dry weight of the *tapetum* [44]. When comparing the results with the literature, it is important to consider that the specialized retina lipids in the joint RTL samples will be diluted by the fibrous *tapetum* tissue. 

Studies with rodents demonstrated that the FA composition of the retina is influenced by diet [45,46,47,48,49,50,51] and, for a given species, the retinal FA composition of each phospholipid class is comparable to that of the brain grey matter [6,52]. The retina of sheep and cattle have a similar FA composition [53], and the main FA of bovine retina (% dry weight) are 16:0 (25%), 18:0 (17%), 18:1 (17%) and DHA (23%) [6]. In the present study, 16:0 averaged 20%, 18:0 averaged 18%, *c*9-18:1 averaged 26% and DHA averaged 6.4%. AA averaged 6%, in line with it being the most abundant omega-6 PUFA in the retina [7,8]. 

Contrary to what was observed in the brain, EPA supplementation increased the EPA content in the RTL tissues. In the C lambs, EPA averaged 0.18% TFA + DMA, which was in line with human EPA retinal content [54]. The EPA content significantly increased in O-fed lambs (0.59% TFA + DMA) and particularly in SD- and FD-fed lambs (0.82% TFA + DMA; + 4.6 times the EPA content in the C-fed lambs). Contrary to the results achieved in the brain, our results showed that the RTL tissues of lambs are very responsive to EPA supplementation. The same magnitude of response was only comparable to what we found in the liver (Figure 4). 

The 4.6-fold increase was achieved despite the short duration of EPA supplementation. Consistent with a better responsiveness of RTL tissues to the experimental diets, control lambs were clearly separated from the lambs consuming *Nannochloropsis*-supplemented diets in the sPLSDA analysis. This shows, once again, that the RTL tissues seem to have been much more sensitive to dietary intervention. The high responsiveness of the retina is evident in rodent studies, in which EPA contents of 6 to 35 times greater have been reported following EPA supplementation [26,55].

As in the brain, no alterations in DHA content were observed between different treatments. The content of DHA in RTL tissues averaged 6.4% of the total TFA + DMA. This is considerably lower that what was reported in previous studies for ruminants in which the DHA content averaged approximately 20–30% [6,53,56].

## 5. Conclusions

After a short-term trail of EPA supplementation in lambs, achieved through feeding using three different diets containing *Nannochloropsis* sp. microalga, it was possible to conclude that the brain content of EPA was not responsive to dietary supplementation. However, the EPA content in the retina was highly responsive in lambs supplemented with *Nannochloropsis*, especially lambs consuming SD and FD diets. Although we could not confirm an advantage in freeze-drying over spray-drying *Nannochloropsis oceanica* with respect to the efficiency of EPA enhancement in the lambs’ retinal tissues, we can confirm their advantage over the free oil. Overall, our results suggest that RTL is a good target to evaluate the differences in n-3 LC-PUFA absorption due to the changes in rumen biohydrogenation associated with dietary interventions.

## Figures and Tables

**Figure 1 animals-13-00828-f001:**
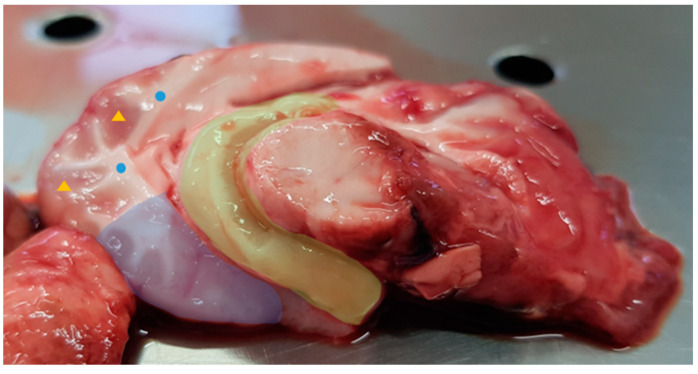
Lamb right brain hemisphere. The four collected parts are highlighted in the figure: prefrontal cortex and hippocampus are filled in blue and yellow, respectively; grey matter is identified with the yellow triangles and white matter with the blue circles.

**Figure 2 animals-13-00828-f002:**
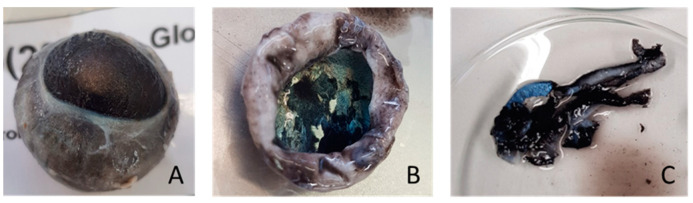
Dissection of the right eyeball of one lamb. From left to right: (**A**) right eyeball; (**B**) removal of eye structures (cornea, iris and lens) and *tapetum lucidum* evidenced; and (**C**) cutting of material used for FA analysis including *tapetum lucidum* and retina tissues.

**Figure 3 animals-13-00828-f003:**
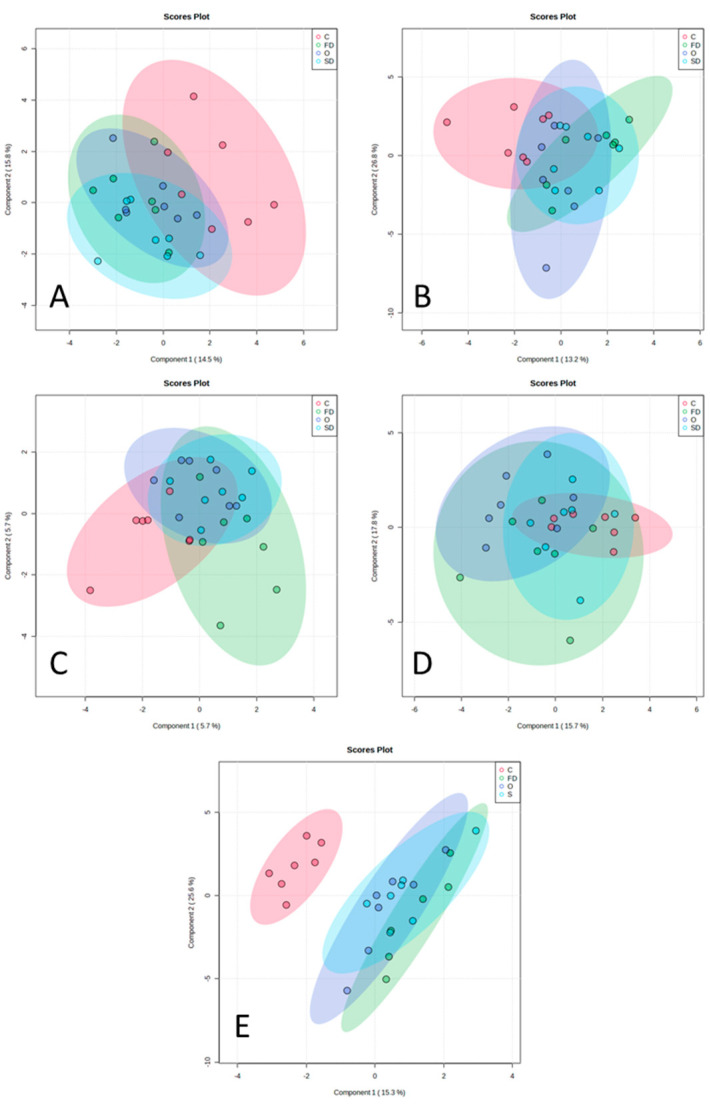
sPLSDA analysis including the FA + DMA profile. The different plots correspond to the different brain parts (**A**–**D**) and RTL tissues (**E**). (**A**) prefrontal cortex, (**B**) hippocampus, (**C**) grey matter, (**D**) white matter, and (**E**): RTL tissues. Diets are represented in different colours: red—C, control diet with no EPA sources; purple—O, diet with *Nannochloropsis* sp. oil; blue—SD, diet with spray-dried *Nannochloropsis oceanica* biomass; and green—FD, diet with freeze-dried *Nannochloropsis oceanica* biomass.

**Figure 4 animals-13-00828-f004:**
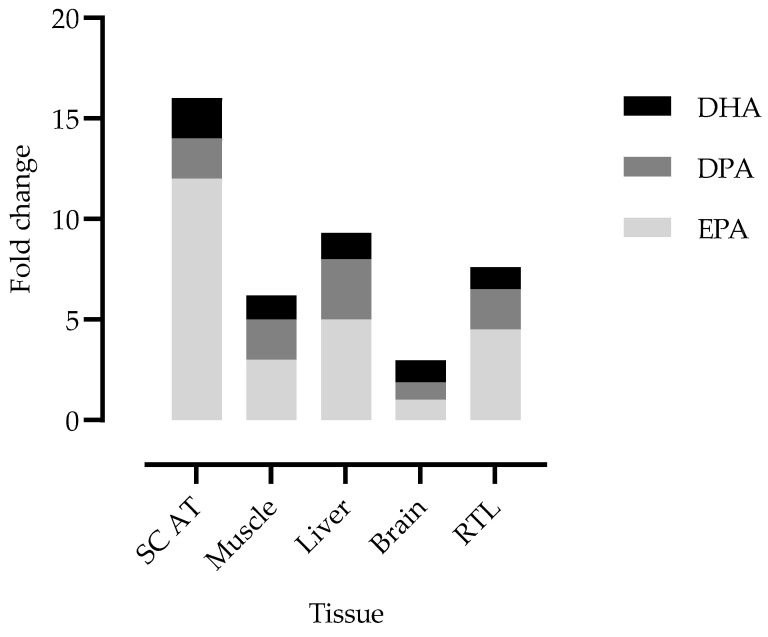
EPA, DPA and DHA fold change in lambs’ tissues. The fold change was calculated between the mean value determined in the tissues of freeze-dried *Nannochloropsis oceanica*-fed lambs versus the mean value for control-fed lambs. SC AT: subcutaneous adipose tissue; RTL: retina and *tapetum lucidum*. The reference value for the brain corresponds to the mean of all brain parts.

**Table 1 animals-13-00828-t001:** Total fatty acid content (g/kg dry matter) and fatty acids (FAs) profile (% of total fatty acids) of the experimental diets.

Item	Diets ^2^			
	C	O	SD	FD
Total FA ^1^	13.7	20.6	19.5	20.2
FA profile				
14:0	n.d.	1.26	2.40	2.28
16:0	25.5	23	23.9	26.2
*c*9-16:1	0.51	5.15	10.2	8.22
17:0	0.66	0.34	n.d.	n.d.
18:0	4.09	2.96	2.51	3.32
*c*9-18:1	18.6	14.5	11.7	13.7
*c*11-18:1	0.73	0.93	0.72	0.69
18:2n-6	41	31.5	28.2	29.1
18:3n-3	8.91	6.80	5.02	6.44
20:4n-6	n.d	2.58	3.69	3.17
20:5n-3	n.d.	10.40	11.3	6.88
22:0	n.d.	0.58	0.36	n.d.

^1^ FA—fatty acids. ^2^ C—control diet with no EPA sources; O—diet with *Nannochloropsis* sp. oil; SD—diet with spray-dried *Nannochloropsis oceanica* biomass; FD—diet with freeze-dried *Nannochloropsis oceanica* biomas; n.d.—not detected. In the FA notation (x:n-), ‘x’ represents the number of C atoms, ‘:’ the number of double bonds and ‘n-’ the location, in its carbon chain, of the double bond which is closest to the methyl end of the molecule. *c* stands for *cis*. Adapted from [16].

**Table 2 animals-13-00828-t002:** Daily fatty acid (FA) intake (m/day) during the trial.

FA ^1^	Diets ^2^				SEM ^3^	*p*-Value C vs. *Nannochloropsis* DIETS ^4^
	C	O	SD	FD		
14:0	-	320	590	540	0.030	<0.001
16:0	4110	5780	5710	6200	0.466	0.217
*c*9-16:1	60	1300	2490	1970	0.119	<0.001
18:0	660	740	600	790	0.061	0.011
*c*9-18:1	3010	3620	2800	3260	0.277	<0.001
*c*11-18:1	120	230	110	160	0.014	<0.001
18:2n-6	6640	7900	6710	6910	0.607	<0.001
18:3n-3	1440	1700	1190	1530	0.129	<0.001
20:4n-6	-	670	890	750	0.058	<0.001
20:5n-3	-	2600	2710	1630	0.178	<0.001
Total	16,100	25,100	24,000	23,700	1.90	0.033

^1^ FA—fatty acid; ^2^ C— control diet with no EPA sources; O—diet with *Nannochloropsis* sp. oil; SD—diet with spray-dried *Nannochloropsis oceanica* biomass; FD—diet with freeze-dried *Nannochloropsis oceanica* biomass; n.d.—not detected. ^3^ Standard error of the mean. ^4^ C vs. *Nannochloropsis* diets, compares C with O, SD and FD together. In the FA notation (x:n-), ‘x’ represents the number of C atoms, ‘:’ the number of double bonds and ‘n-’ the location, in its carbon chain, of the double bond which is closest to the methyl end of the molecule. *c* stands for *cis*.

**Table 3 animals-13-00828-t003:** Total fatty acid (TFA) and dimethyl acetal (DMA) content (mg/g DM) and composition (% TFA + DMA) of the grey matter and white matter of lambs.

FA and DMA ^1^	Grey Matter						White Matter					
	Diets ^2^				SEM ^3^	*p*-Value	Diets ^2^				SEM ^3^	*p*-Value
	C	O	SD	FD			C	O	SD	FD		
TFA + DMA	189	192	190	193	5.2	0.970	235	227	225	222	4.1	0.163
14:0	0.57	0.55	0.58	0.58	0.025	0.746	0.52	0.52	0.55	0.53	0.018	0.778
15:0	0.11	0.12	0.11	0.13	0.008	0.392	0.09	0.09	0.09	0.11	0.007	0.177
16:0	19	19	19	19	0.4	0.942	13	14	14	15	0.4	0.261
*c*7-16:1	0.42	0.42	0.40	0.42	0.021	0.934	0.35	0.37	0.35	0.36	0.016	0.722
*c*9-16:1	0.42	0.43	0.45	0.45	0.024	0.810	0.26	0.29	0.27	0.29	0.013	0.354
17:0	0.29	0.29	0.27	0.31	0.014	0.445	0.27	0.31	0.28	0.31	0.015	0.181
*c*9-17:1	0.11	0.11	0.10	0.11	0.010	0.940	0.11	0.17	0.14	0.15	0.019	0.223
18:0	21	21	20	21	0.5	0.964	15	16	16	16	0.4	0.176
*c*9-18:1	15	15	15	14	0.5	0.847	22	22	22	21	0.5	0.216
*c*11-18:1	3.4	3.3	3.3	3.3	0.07	0.433	3.0	3.0	2.9	3.0	0.05	0.671
18:2n-6	0.48	0.48	0.50	0.52	0.030	0.673	0.29	0.43	0.39	0.42	0.042	0.111
19:1	0.06	0.07	0.07	0.07	0.006	0.836	0.10	0.10	0.10	0.10	0.008	0.892
20:0	0.32	0.31	0.31	0.32	0.017	0.870	0.69 ^a^	0.64 ^ab^	0.62 ^b^	0.62 ^b^	0.019	0.045
18:3n-3	0.04	0.03	0.03	0.04	0.008	0.843	0.13 ^a^	0.08 ^b^	0.13 ^a^	0.09 ^b^	0.013	0.026
*c*11-20:1	0.86	0.83	0.88	0.80	0.080	0.902	2.2	2.0	2.1	1.9	0.11	0.372
20:1	0.27	0.27	0.27	0.26	0.021	0.941	0.62	0.62	0.59	0.56	0.035	0.542
21:0	0.06	0.06	0.07	0.07	0.005	0.428	0.09	0.10	0.11	0.10	0.012	0.868
20:2	0.10	0.12	0.11	0.12	0.014	0.761	0.26	0.25	0.24	0.23	0.023	0.803
20:2n-6	0.10	0.11	0.11	0.10	0.019	0.959	0.15	0.14	0.14	0.14	0.022	0.638
20:3n-9	0.54	0.54	0.59	0.49	0.068	0.764	0.69	0.69	0.69	0.65	0.054	0.943
22:0	0.71	0.66	0.64	0.66	0.069	0.915	1.90	1.84	1.68	1.69	0.078	0.140
20:3n-6	0.33	0.39	0.36	0.39	0.020	0.094	0.35	0.42	0.38	0.45	0.030	0.111
22:1	0.33	0.32	0.31	0.30	0.040	0.970	0.80	0.10	0.40	0.37	0.170	0.068
20:3n-3	0.13	0.13	0.12	0.11	0.020	0.941	0.57	0.78	0.62	0.60	0.083	0.334
22:1/20:3n-3	0.28	0.20	0.30	0.27	0.064	0.727	0.81	0.88	0.98	0.70	0.067	0.154
20:4n-6	6.3	5.8	5.7	6.2	0.24	0.312	3.8	4.0	3.8	4.2	0.16	0.176
23:0	0.21	0.20	0.21	0.20	0.028	0.992	0.50	0.39	0.39	0.37	0.050	0.262
22:2n-6	0.09	0.09	0.09	0.09	0.018	0.985	0.52	0.48	0.49	0.60	0.119	0.863
20:5n-3	0.26	0.29	0.32	0.32	0.040	0.655	0.71	0.60	0.69	0.69	0.053	0.502
24:0	0.66	0.63	0.61	0.63	0.139	0.997	2.3	2.3	2.2	2.0	0.11	0.115
*c*15-24:1	1.2	1.3	1.4	1.3	0.20	0.951	4.4	3.9	4.2	3.7	0.25	0.251
24:1	0.69	0.20	0.23	0.22	0.200	0.284	0.70	0.65	0.65	0.63	0.038	0.576
22:4n-6	2.5	2.5	2.6	2.9	0.268	0.864	2.8	2.4	2.5	2.7	0.14	0.194
21:5	0.11	0.03	0.03	0.03	0.035	0.334	0.14	0.17	0.12	0.10	0.030	0.448
22:5n-6	0.62	0.41	0.38	0.38	0.066	0.060	0.34	0.25	0.25	0.25	0.037	0.251
26:0	0.15	0.02	0.03	0.04	0.053	0.304	0.08	0.07	0.08	0.08	0.012	0.892
26:1	0.08	0.10	0.13	0.09	0.049	0.878	0.26	0.34	0.42	0.23	0.036	0.069
22:5n-3	3.0	0.76	1.05	1.21	1.111	0.504	0.51	0.32	0.57	0.65	0.150	0.454
22:6n-3	11	14	14	13	1.07	0.248	4.7	5.2	5.2	5.9	0.40	0.212
DMA												
DMA 16:0	2.2	2.2	2.3	2.2	0.10	0.911	3.7	3.6	3.7	3.5	0.09	0.644
DMA 17:0	0.19	0.16	0.16	0.19	0.013	0.320	0.23	0.24	0.23	0.26	0.012	0.319
DMA 18:0	3.9	4.7	4.7	4.7	0.35	0.275	4.5	4.6	4.6	4.7	0.09	0.543
DMA *c*9-18:1	0.92	1.01	1.11	1.01	0.128	0.784	2.8	2.5	2.3	2.5	0.24	0.458
DMA *c*11-18:1	0.95	0.96	1.00	0.95	0.071	0.941	2.0	1.9	2.0	1.9	0.075	0.645
Partial sums												
C18 ^4^	39	39	39	39	0.3	0.526	40	41	41	40	0.5	0.207
C18:1 ^5^	20	0.27	0.27	0.27	9.97	0.418	0.32	0.34	0.33	0.28	0.027	0.355
DMA	15	9.0	9.2	9.1	2.82	0.341	13	13	13	13	0.2	0.596
SFA ^6^	38	42	42	42	1.6	0.270	35	36	35	36	0.6	0.207
MUFA ^7^	22	22	22	22	0.8	0.901	35	34	34	32	0.8	0.156
*cis*-MUFA	22	21	21	21	0.9	0.455	32	32	32	30	0.7	0.223
PUFA ^8^	24	26	26	26	0.9	0.519	16	16	16	18	0.6	0.159
n-3 PUFA	14	15	15	15	0.5	0.121	6.7	7.0	7.3	8.0	0.41	0.162
n-6 PUFA	12	9.8	9.8	10	0.55	0.101	8.1	8.0	7.8	8.7	0.33	0.320
EPA + DHA	18	14	14	14	2.2	0.414	5.4	5.8	5.9	6.6	0.41	0.232
AA/EPA ratio	21	23	19	21	3.07	0.815	5.7	6.9	5.7	6.2	0.56	0.375

Means within a row with different letters are significantly different (*p*  <  0.05). ^1^ FA and DMA—fatty acids and dimethyl acetals; ^2^ C—control diet with no EPA sources; O—diet with *Nannochloropsis* sp. oil; SD—diet with spray-dried *Nannochloropsis oceanica* biomass; FD—diet with freeze-dried *Nannochloropsis oceanica* biomass. ^3^ Standard error of the mean; the value presented corresponds to a pooled sample standard error of the mean. ^4^ Sum of C18 FA. ^5^ Sum of C18:1 FA. ^6^ Sum of saturated FA. ^7^ Sum of monounsaturated FA. ^8^ Sum of polyunsaturated FA. In the FA notation (x:n-), ‘x’ represents the number of C atoms, ‘:’ the number of double bonds and ‘n-’ the location, in its carbon chain, of the double bond which is closest to the methyl end of the molecule. *c* stands for *cis*.

**Table 4 animals-13-00828-t004:** Total fatty acid (TFA) and dimethyl acetal (DMA) (mg/g DM) content and composition (% TFA + DMA) of the hippocampus and prefrontal cortex of lambs.

FA and DMA ^1^	Hippocampus						Prefrontal Cortex					
	Diets ^2^				SEM ^3^	*p*-Value	Diets ^2^				SEM ^3^	*p*-Value
	C	O	SD	FD			C	O	SD	FD		
TFA + DMA	199	196	204	195	4.4	0.569	206	206	211	210	3.0	0.533
14:0	0.52	0.51	0.53	0.48	0.018	0.243	0.59	0.59	0.61	0.63	0.024	0.643
15:0	0.11	0.10	0.09	0.10	0.005	0.185	0.11	0.11	0.10	0.12	0.008	0.493
16:0	17	17	16	16	0.4	0.458	18	17	17	18	0.3	0.923
*c*7-16:1	0.43	0.43	0.40	0.38	0.017	0.101	0.43	0.42	0.40	0.41	0.018	0.649
*c*9-16:1	0.35	0.39	0.38	0.34	0.017	0.174	0.42	0.42	0.41	0.43	0.019	0.863
17:0	0.29	0.28	0.27	0.29	0.009	0.428	0.27	0.29	0.26	0.31	0.019	0.408
*c*9-17:1	0.11	0.12	0.12	0.12	0.009	0.937	0.13	0.13	0.13	0.15	0.010	0.263
18:0	18	19	18	18	0.4	0.642	19	19	19	19	0.4	0.951
*c*9-18:1	17	17	18	17	0.5	0.756	17	17	17	17	0.5	0.905
*c*11-18:1	3.4	3.3	3.2	3.3	0.05	0.325	3.4	3.3	3.2	3.2	0.05	0.107
18:2n-6	0.41	0.45	0.45	0.44	0.027	0.644	0.40	0.42	0.42	0.45	0.035	0.790
19:1	0.08	0.08	0.08	0.09	0.008	0.578	0.08	0.08	0.07	0.08	0.005	0.125
20:0	0.50	0.48	0.49	0.50	0.029	0.968	0.45	0.48	0.44	0.45	0.027	0.705
18:3n-3	0.09	0.08	0.09	0.06	0.013	0.481	0.07	0.08	0.06	0.06	0.008	0.443
*c*11-20:1	1.4	1.3	1.6	1.6	0.14	0.567	1.2	1.2	1.2	1.2	0.07	0.912
20:1	0.38	0.37	0.40	0.39	0.026	0.822	0.38	0.39	0.38	0.37	0.021	0.878
21:0	0.07	0.07	0.08	0.07	0.007	0.904	0.10	0.09	0.08	0.07	0.013	0.660
20:2	0.19	0.17	0.19	0.17	0.012	0.472	0.19	0.20	0.19	0.18	0.018	0.846
20:2n-6	0.12	0.12	0.12	0.15	0.013	0.297	0.08	0.11	0.10	0.11	0.007	0.075
20:3n-9	0.83	0.84	0.77	0.70	0.051	0.221	0.73	0.74	0.69	0.64	0.066	0.673
22:0	1.0	1.1	1.2	1.2	0.12	0.815	1.1	1.2	1.1	1.2	0.07	0.759
20:3n-6	0.37 ^b^	0.44 ^a^	0.42 ^ab^	0.46 ^a^	0.023	0.049	0.29 ^b^	0.38 ^a^	0.35 ^a^	0.37 ^a^	0.018	0.008
22:1	0.50	0.47	0.58	0.59	0.054	0.348	0.46	0.53	0.48	0.48	0.032	0.536
20:3n-3	0.24	0.21	0.25	0.24	0.022	0.588	0.26	0.29	0.25	0.27	0.022	0.572
22:1/20:3n-3	0.59	0.49	0.55	0.57	0.145	0.960	0.43	0.30	0.46	0.46	0.037	0.108
20:4n-6	5.4	5.4	4.9	5.1	0.26	0.476	5.3	5.0	4.9	5.1	0.17	0.346
23:0	0.38	0.33	0.36	0.37	0.030	0.641	0.24	0.25	0.24	0.25	0.022	0.980
22:2n-6	0.18	0.13	0.19	0.19	0.035	0.600	0.18	0.20	0.21	0.20	0.038	0.958
20:5n-3	0.56	0.59	0.63	0.62	0.035	0.580	0.55	0.65	0.63	0.65	0.061	0.596
24:0	1.4	1.4	1.5	1.5	0.14	0.971	1.1	1.3	1.2	1.2	0.10	0.805
*c*15-24:1	2.6	2.1	2.7	2.8	0.28	0.266	2.2	2.3	2.3	2.3	0.17	0.983
24:1	0.39	0.31	0.38	0.40	0.034	0.260	0.43	0.42	0.40	0.44	0.030	0.823
22:4n-6	3.2	2.8	2.8	2.9	0.16	0.235	2.8	2.5	2.6	2.6	0.13	0.329
21:5	0.09	0.08	0.09	0.10	0.014	0.861	0.06	0.07	0.07	0.06	0.008	0.413
22:5n-6	0.50 ^a^	0.34 ^b^	0.31 ^b^	0.31 ^b^	0.052	0.047	0.52 ^a^	0.38 ^ab^	0.34 ^b^	0.33 ^b^	0.052	0.056
26:0	0.05	0.04	0.05	0.05	0.005	0.286	0.05	0.05	0.05	0.05	0.005	0.675
26:1	0.26	0.18	0.23	0.25	0.067	0.846	0.52	0.35	0.60	0.57	0.078	0.256
22:5n-3	0.83 ^c^	1.14 ^b^	1.24 ^ab^	1.36 ^a^	0.066	<0.001	0.54 ^c^	0.82 ^b^	0.97 ^ab^	1.06 ^a^	0.069	<0.001
22:6n-3	9.0	9.6	8.9	9.0	0.38	0.573	10	10	10	10	0.53	0.956
DMA												
DMA 16:0	2.7	2.7	2.8	2.7	0.09	0.717	2.7	2.9	2.8	2.8	0.12	0.910
DMA 17:0	0.20	0.20	0.20	0.22	0.013	0.716	0.19	0.19	0.19	0.21	0.013	0.373
DMA 18:0	4.6	4.7	4.7	4.8	0.08	0.612	4.5	4.7	4.7	4.6	0.09	0.519
DMA *c*9-18:1	1.4	1.5	1.8	1.8	0.17	0.268	1.5	1.6	1.6	1.6	0.11	0.986
DMA *c*11-18:1	1.4	1.4	1.5	1.4	0.05	0.301	1.3	1.4	1.4	1.4	0.08	0.919
Partial sums												
C18 ^4^	40	40	40	39	0.3	0.303	40	40	40	40	0.3	0.813
C18:1 ^5^	0.37	0.32	0.33	0.33	0.018	0.265	0.29	0.28	0.26	0.27	0.019	0.807
DMA	10.3	10	11	11	0.3	0.320	10	11	11	11	0.3	0.830
SFA ^6^	39	40	39	39	0.6	0.311	40	40	40	40	0.5	0.934
MUFA ^7^	27	26	28	27	0.9	0.617	27	27	27	27	0.7	0.979
*cis*-MUFA	25	25	26	25	0.8	0.671	25	25	25	25	0.6	0.972
PUFA ^8^	22	22	21	22	0.7	0.744	22	22	22	22	0.6	0.952
n-3 PUFA	11	12	11	11	0.3	0.340	12	12	12	12	0.5	0.790
n-6 PUFA	10	9.7	9.2	9.6	0.41	0.362	9.6	9.0	8.9	9.2	0.29	0.309
EPA + DHA	9.6	10	9.5	9.6	0.39	0.600	10	11.0	11	11	0.5	0.960
AA/EPA ratio	9.8	9.2	8.1	8.4	0.72	0.366	10	8.4	8.2	8.3	1.13	0.476

Means within a row with different letters are significantly different (*p*  <  0.05). ^1^ FA and DMA—fatty acids and dimethyl acetals; ^2^ C—control diet with no EPA sources; O—diet with *Nannochloropsis* sp. oil; SD—diet with spray-dried *Nannochloropsis oceanica* biomass; FD—diet with freeze-dried *Nannochloropsis oceanica* biomass. ^3^ Standard error of the mean; the value presented corresponds to a pooled sample standard error of the mean. ^4^ Sum of C18 FA. ^5^ Sum of C18:1 FA. ^6^ Sum of saturated FA. ^7^ Sum of monounsaturated FA. ^8^ Sum of polyunsaturated FA. In the FA notation (x:n-), ‘x’ represents the number of C atoms, ‘:’ the number of double bonds and ‘n-’ the location, in its carbon chain, of the double bond which is closest to the methyl end of the molecule. *c* stands for *cis*.

**Table 5 animals-13-00828-t005:** Total fatty acid (TFA) and dimethyl acetal (DMA) (mg/g DM) content and composition (%TFA + DMA) of the RTL tissues of lambs.

FA and DMA ^1^	Diet ^2^				SEM ^3^	*p*-Value
	C	O	SD	FD		
TFA + DMA	54	61	43	53	8.9	0.591
10:0	0.05	0.06	0.08	0.06	0.014	0.376
12:0	0.26	0.24	0.23	0.25	0.036	0.955
14:0	2.2	1.3	1.6	1.9	0.25	0.069
*i*-15:0	0.08	0.05	0.05	0.02	0.014	0.063
*a*-15:0	0.18	0.06	0.07	0.05	0.066	0.433
*c*9-14:1	0.06	0.08	0.09	0.07	0.017	0.518
15:0	0.27	0.27	0.28	0.31	0.026	0.766
*i*-16:0	0.09	0.07	0.10	0.06	0.018	0.341
16:0	21	21	20	21	0.58	0.608
*i*-17:0	0.15	0.17	0.17	0.15	0.027	0.881
*c*7-16:1	0.29	0.30	0.25	0.29	0.034	0.717
*c*9-16:1	0.77	0.77	0.66	0.83	0.118	0.752
*a*-17:0	0.18	0.14	0.09	0.13	0.041	0.488
17:0	0.67	0.70	0.60	0.74	0.038	0.084
*c*9-17:1	0.31	0.32	0.27	0.27	0.033	0.543
18:0	18	18	19	18	0.5	0.591
*t*6/*t*7/*t*8-18:1	0.08	0.11	0.11	0.09	0.019	0.664
*t*9-18:1	0.07	0.12	0.11	0.11	0.018	0.198
*t*10-18:1	0.49	0.53	0.28	0.35	0.096	0.219
*t*11-18:1	0.66	0.87	0.88	0.87	0.114	0.455
*t*12-18:1	0.20	0.22	3.56	0.23	1.784	0.437
*c*9-18:1	29	28	21	27	2.4	0.121
*c*11-18:1	1.8	1.8	2.0	1.9	0.10	0.415
*c*12-18:1	0.09	0.11	0.15	0.10	0.023	0.241
*c*13-18:1	0.06	0.06	0.04	0.05	0.014	0.728
*t*16/*c*14-18:1	0.09	0.13	0.10	0.12	0.025	0.671
*c*15-18:1	0.04	0.04	0.05	0.03	0.014	0.799
*tc*/*ct*-18:2/cyclo-17	0.11	0.18	0.16	0.22	0.043	0.370
*c*9,*t*12/*c*9,*t*15/*t*8,*c*13-18:2	0.09	0.10	0.07	0.11	0.021	0.626
*c*16-18:1	0.03 ^b^	0.03 ^b^	0.08 ^a^	0.04 ^b^	0.012	0.021
*t*9,*c*12-18:2	0.04	0.06	0.07	0.05	0.013	0.385
*t*11,*c*15/*t*10,*c*15-18:2	0.06	0.16	0.08	0.14	0.033	0.094
18:2n-6	0.31	3.75	4.25	3.98	0.299	0.317
20:0	0.16	0.17	0.20	0.17	0.022	0.675
18:3n-3	0.63	0.67	0.65	0.66	0.038	0.873
*c*9,*t*11-CLA	0.26	0.37	0.30	0.38	0.057	0.339
20:2n-6	0.15	0.16	0.19	0.17	0.020	0.509
20:3n-9	0.29 ^a^	0.20 ^bc^	0.24 ^ab^	0.14 ^c^	0.032	0.014
22:0	0.05	0.07	0.07	0.06	0.012	0.414
20:3n-6	0.38 ^b^	0.59 ^a^	0.70 ^ab^	0.58 ^a^	0.056	0.004
20:3n-3	0.18	0.22	0.22	0.20	0.029	0.646
20:4n-6	6.4	5.4	6.5	5.6	0.54	0.391
20:5n-3	0.18 ^c^	0.59 ^b^	0.83 ^a^	0.81 ^a^	0.057	<0.001
22:4n-6	0.93 ^a^	0.62 ^b^	0.73 ^ab^	0.58 ^b^	0.071	0.008
22:5n-3	1.2 ^c^	2.0 ^b^	2.5 ^a^	2.4 ^ab^	0.17	<0.001
22:6n-3	5.7	6.8	6.9	6.2	0.90	0.752
DMA						
DMA 16:0	1.2	1.2	1.5	1.2	0.14	0.195
DMA 18:0	1.8	1.9	2.1	1.7	0.18	0.552
DMA 18:1	0.41	0.37	0.43	0.34	0.041	0.360
Partial sums						
C18 ^4^	37	37	34	36	1.6	0.485
DMA	3.4	3.4	4.0	3.3	0.3	0.351
SFA ^5^	42	41	41	42	0.40	0.417
MUFA ^6^	34	33	30	32	1.7	0.330
*cis*-MUFA	32	31	25	30	2.4	0.141
PUFA ^7^	20	22	24	22	1.6	0.266
n3-PUFA	7.9	10	11	10	1.0	0.130
n6-PUFA	11	11	12	11	0.9	0.472
EPA + DHA	5.9	7.4	7.7	7.0	0.91	0.491
AA/EPA ratio	37 ^a^	9.1 ^b^	8.0 ^b^	6.9 ^b^	2.33	<0.001

Means within a row with different letters are significantly different (*p*  <  0.05). ^1^ FA and DMA—fatty acids and dimethyl acetals; ^2^ C—control diet with no EPA sources; O—diet with *Nannochloropsis* sp. oil; SD—diet with spray-dried *Nannochloropsis oceanica* biomass; FD—diet with freeze-dried *Nannochloropsis oceanica* biomass. ^3^ Standard error of the mean; the value presented corresponds to a pooled sample standard error of the mean. ^4^ Sum of C18 FA. ^5^ Sum of saturated FA. ^6^ Sum of monounsaturated FA. ^7^ Sum of polyunsaturated FA. In the FA notation (x:n-), ‘x’ represents the number of C atoms, ‘:’ the number of double bonds and ‘n-’ the location, in its carbon chain, of the double bond which is closest to the methyl end of the molecule. *c* stands for *cis* and *t* stands for *trans*. *i* stands for *iso* and *a* stands for *anteiso*.

## Data Availability

The data presented in this study are available on request from the corresponding author.

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
