# Peer review of "Enrichment of Brain n-3 Docosapentaenoic Acid (DPA) and Retinal n-3 Eicosapentaenoic Acid (EPA) in Lambs Fed Nannochloropsis oceanica Microalga"

_animals, 2023, doi:10.3390/ani13050828_

Round 1
Reviewer 1 Report
General comments:
This manuscript attempts to describe the effects of very short term feeding of a microalgal source of EPA and DHA on brain and retinal fatty acid composition. What appears to be missing are a rationale for the short term of supplementation, a table showing the composition of the diets, especially the fatty acid composition and content of the algal lipids, intake of fatty acids, and evidence that the long-chain polyunsaturated fatty acids had been absorbed. Authors indicated that muscle fatty acid composition of lambs fed microalgae was responsive to such treatments, and that they have yet unpublished data showing that liver fatty acids were responsive to microalgae supplementation. If the muscle fatty acid results that were cited, as well as the liver fatty acid data mentioned in the discussion, were from the same lambs, either a table or a summary showing relevant results would be necessary to strengthen their conclusions. The liver data would be especially valuable because there is ample evidence in the published literature that the primary source of long-chain PUFA for brain lipids is the liver.
Specific comments and suggestions:
Line Comment
27 …”scarce data are”… “data” is always plural.
29 Change to …”can selectively concentrate some n-3 LC-PUFA…” C18:3 n-3 and EPA are quite low in brain.
36 Suggest change to “We conclude that retinal tissues are sensitive to short-term n-3 PUFA supplementation in lambs.”
38 Key Words should include words not in the title. For example: Fatty acid; Hippocampus; Prefrontal Cortex; Dimethyl Acetal.
58 Delete the word …”also”… in first sentence.
59 Change …”retina’s”… to …”retinal”…
67-68 Awkward sentence. Delete or re-write. Do authors suggest that ovine brain fatty acid composition is assumed to be similar to that of bovine brain because both are ruminants? As referenced by #10?
68 Change …”lipids composition” to …”fatty acid composition”
69-70 Evidence suggests that brain, and likely retinal, DHA and AA would arise from the liver stores, not the diet. Author should review some additional literature on this aspect. Suggestions: Chen, C.T., et al, 2008, Prost. Leuk. And Essent Fatty Acid, 79:85-91; and Rapoport et al., 2010, Prost. Leuk. Essent Fatty Acids, 82:273-276.
78-81 Awkward sentence. The reference to #14 is the author’s publication reporting the protective effect of drying method on biohydrogenation of PUFA. For the current report, it’s not necessary to reference this in the objectives statement. Indicate that the purpose is to determine effects of drying method on FA in brain and retinal tissue.
89 What were the pen size dimensions? Also, delete the word “different.”
89-90 Spell out “4” and “7”. Numeral designations should only be used to precede quantitative measures.
90-94 Streamline this sentence. Suggest “Experimental diets included…..” then describe them.
At this point authors indicated a 21-day feeding period. Indicate the rationale for this length of time fed the experimental diets because it seems quite short to affect fatty acids in the brain. Provide references to strengthen rationale for the short period.
98 Feed intake results should not be in the Methods section. Move to the Results section. This is a vital part of the paper because readers will need to know what the diet composition was, the fatty acid composition and content of the diet, and comparison of feed intake by treatment. Authors only indicated averages, which does not describe the impact of the various treatments.
114 Define “RTL”
125-130 Be specific on volume of HCl/Methanol, and concentration of HC/ in the methanol. The amount of total lipids left as un-transesterifed by allowing the sample to sit in the methanolic HCl after an overnight stand can be significant. Comment on how quantitative the conversion of total fatty acid to FAME was. There should have been no evidence of a lipid droplet in the HCl/Methanol tube if conversion was quantitative.
Authors allude to the genesis of the dimethyl acetal in the Discussion section. That part should be included here in the methods section.
140 Was the same column used for the MS analysis that was used for the GC-FID analysis?
153-154 Units for fatty acid composition are not clearly stated. Should be g of FA /100 g of total FA or / 100 g of total FA+DMA.
Tables 1 and 2. Each table should stand alone and contain its own footnote. The lines in the tables are not drawn correctly (see data for TFA + DMA). The footnotes need to describe more than just the treatments. Indicate what the fatty acid designations represent. For example, # : # is number of carbon atoms : number of carbon-carbon double bonds. Also, indicate “n” with SEM in the footnote.
168-180 This section could be shortened considerably. One statement is needed for white matter and grey matter. For example, “No microalgal supplementation effects (P > 0.05) were observed for any of the FA and DMA.” Reporting the averages of for specific FA is of little use. More interesting would have been the comparison of white and grey matter FA composition. Do not repeat in text the data that are within the table.
179-180 Where are the data for “total C20” in the tables? What is a “small statistically significant change” mean? P-value?
Throughout the results there are grammatical issues that need attention.
Table 3. Same issues as indicated for Tables 1 and 2.
In addition, more descriptions of fatty acid designations are needed in this table. For example, i-16:0 and a-16:0. Not all readers will understand the “i” and “a” as “iso” and “anti-iso” or what it means. These need to be defined. Also, “c” and “t” need to be defined or indicated for the fatty acid directly as cis or trans within the table.
212-213 Delete first part of the sentence. Start with “content of TFA + DMA was not affected (P > 0.05) by dietary treatment.”
213-215 Delete this sentence. Not needed.
225 As an example of improved presentation to use throughout the paper, in the sentence delete …”significantly differed between treatments”… and state as: “c16-18:1 was greater (P < 0.05) in SD lambs than all other treatments.” And do not show data in the sentence that are in the table.
234-235 Delete the sentence. Not needed.
242-244 Delete. No statistical contrasts were conducted to compare brain with RTL tissues.
345 Assume “DW” is dry weight? It’s only used twice so just spell it out.
359 …”+4.6 times de EPA”… ???
360 Replace “data” with “results”
363-366 Re-write this sentence. Very confusing.
370-374 Very awkward sentence. Re-write and streamline to be more direct.
Author Response
"Please see the attachment."

Reviewer 2 Report
The manuscript presents a correctly designed and analyzed study with novel and relevant results.
The only suggestion that I could make to the authors is to address in the discussion how the presentation of the microalga, spray-dried or freeze-dried, could affect the results and make a recommendation in this regard in the conclusions. This however does not change my recommendation on the article.
Round 2
Reviewer 1 Report
Second Revision:
This revision was a marked improvement over the first draft. However, I have suggested several additional revisions, primarily with sentence structure and interpretation. These are listed below by line number of the revised manuscript.
13-22 Summary: Include length of time supplements were fed to the lambs. Exactly 21 days??
31 …”supplemented with an EPA-rich microalga fed for 21 days”
36 Instead of …”relevant” EPA enhancement” indicate the fold increase or percentage increase.
41 Change to: …”Fatty acids are the most abundant organic compounds in brain”…
56 Replace “pointed” with “suggested.”
58 Change to: “Retinal tissue also is rich in lipids”…
65-67 Authors need to indicate that, while Christie, and those cited in his reviews, have reported brain fatty acids, they mostly characterized types of lipids and some fatty acids, but primarily in homogenized brain or gross anatomical structures within the brain, but not in specific functional regions or tracts of the ruminant brain.
73 Change “meat” to “muscle”
73-79 Replace with the following: “Liver stores of n-3 LC-FA was reported to be the primary source of these FA for brain tissues of rats (13), even during periods of low dietary intake of these FA (13, 14).
79-83 Upon this second review it has become apparent that no discussion of using microalga for supplementing LC-PUFA was mentioned in the introduction. The statement of hypothesis focuses on the use of microalga to supply these fatty acids. Authors should present a sentence or two or three indicating the efficacy of microalga for ruminants. Their reference #16, could provide this justification. Also, the processing methods should be discussed as well since this is part of the hypothesis. Thus, if preceding their hypothesis with the word “therefore”, authors need to include these changes.
The Methods section excluded liver sampling method and lobe location, as well as analysis, which is needed since authors have added a supplementary table showing liver fatty acids. If the liver samples were subjected to the same procedures as for brain and RTL tissues, just add liver to the list on or about line 138.
174 “3.1.” should now be “Fatty acid intake”
175 Delete “The animal’s.” Start the sentence with “Feed intake”……
Table 2 In the title spell out “FA” and indicate in the footnotes the abbreviations for fatty acids as “number of carbon atoms : number of carbon-carbon double bonds”
After the table, present the results in text form within the manuscript.
184 Start this section as: “3.2. Brain fatty acid and dimethyl acetal profile.”
Tables 3, 4, and S1 (lines 196, 216, and 241) Footnotes. Delete the first sentence because “values” are not “mean ± SEM, they are just the means. The SEM was a pooled SEM.
205 Need a space before the paragraph starts.
224-226 Suggest changing to: “DPA was lowest (P < 0.05) in hippocampus and prefrontal cortex of C lambs, and highest in these tissues of lambs fed the FD supplement. DPA was lower (P < 0.05) in these tissues in O-fed lambs than in SD-fed lambs.”
227-229 Replace the sentence with the following: “Supplement treatments did not affect (P > 0.05) EPA or DHA concentrations in hippocampus or prefrontal cortex.”
233 Change heading to “3.2”
234 Replace “verified” with “observed”…..”FA concentrations”
235-237 Change to: “However, more treatment effects occurred for RTL tissues than were observed in brain tissues.”
Figure 3 The figure should stand alone, thus its title and footnotes should be understandable without the text in the body of the manuscript. Authors should spell out FD and do not use “C”, just its definition, “Control.” “Fold change” is self-explanatory; delete the sentence using “A’ and “B” (Lines 276-277)
310 Replace “transformed in” to “converted to”
317 Replace “due to the fact that” to “because”
326 Replace “none” with “any”
334-335 Delete …….”thus maybe…..longer feeding period.”
339 Replace “verify” with “observed”
350 Replace “trough” with “through”
355 I suggest “even though” should precede ….”DPA”…
391 Replace “cow” with “cattle”
396 Replace “verify” with “observed” and delete “efficiently” then end the sentence at ….”RTL tissue”
398 Replace ….”which is in line” to “which was in line”
402 Because “intestinal absorption” was not reported in this manuscript, shouldn’t …”EPA supplementation” be used instead?
404 Change “4.6 times” to “4.6-fold”
405 Suggest replacing “Probably inline” with “consistent”
406 Replace …”it was possible to clearly group C lambs separately” with “Control lambs were clearly separated from”…
408 Replace “sensible” with “sensitive”
409 Insert “greater” between “…times EPA”…
411 Replace “verified” with “observed”
413 Replace “is” in both places with “was”
414-415 Delete last sentence.
418 Replace “thorough” with “through”
425 Replace “samples might” to “is”. Have confidence in your results!
Table S1 This is an unusual table, but not necessarily unneeded. However, I highly recommend that it be shortened to show only the fatty acids in the supplements plus DPA and DHA. This table is included to illustrate how dietary fatty acids were digested, absorbed, and deposited in liver.
Table S1 footnote. Change as per the other table footnotes.
